# Fabrication of Nylon-6 and Nylon-11 Nanoplastics and Evaluation in Mammalian Cells

**DOI:** 10.3390/nano12152699

**Published:** 2022-08-05

**Authors:** Sai Archana Krovi, Maria M. Moreno Caffaro, Shyam Aravamudhan, Ninell P. Mortensen, Leah M. Johnson

**Affiliations:** 1RTI International, 3040 E. Cornwallis Drive, Research Triangle Park, Durham, NC 27709, USA; 2Joint School of Nanoscience and Nanoengineering, North Carolina A&T State University, 2907 E. Gate City Blvd., Greensboro, NC 27401, USA

**Keywords:** nylon, nanoplastics, microplastics, nanoplastics fabrication, nanoplastics cellular response

## Abstract

Microplastics (MPs) and nanoplastics (NPs) exist in certain environments, beverages, and food products. However, the ultimate risk and consequences of MPs and NPs on human health remain largely unknown. Studies involving the biological effects of small-scale plastics have predominantly used commercially available polystyrene beads, which cannot represent the breadth of globally dominant plastics. Nylon is a commodity plastic that is used across various industry sectors with substantial global production. Here, a series of well-characterized nylon-11 and nylon-6 NPs were successfully fabricated with size distributions of approximately 100 nm and 500 nm, respectively. The facile fabrication steps enabled the incorporation of fluorescent tracers in these NPs to aid the intracellular tracking of particles. RAW 264.7 macrophages were exposed to nylon NPs in a dose-dependent manner and cytotoxic concentrations and cellular uptake were determined. These well-characterized nylon NPs support future steps to assess how the composition and physicochemical properties may affect complex biological systems and ultimately human health.

## 1. Introduction

The substantial reliance on plastics in modern society has resulted in an estimated worldwide production of plastic materials near 367 million tons in 2020 [1]. It was predicted that 79% of plastics reside in landfills or the environment as of 2015 [2], spurring solutions towards a circular economy of plastics through technical, economic, and social changes [3,4,5]. Despite these efforts, plastics remain ubiquitous, as evident from an emerging insight that small-scale plastics, termed nanoplastics (NPs) and microplastics (MPs), are present within the environment [6,7], foodstuffs [8,9,10,11], beverages [12,13,14], and drinking water [15,16]. The origins of NPs and MPs have been categorized as either primary sources that are intentionally manufactured (e.g., nurdles, microbeads) or as secondary sources that result from the unintentional degradation of macroscale plastic [17]. These diverse origins, combined with varied environmental exposure during the life of the plastic, have resulted in small-scale plastics with a plethora of shapes (e.g., spheres, fibers, fragments), sizes (e.g., macro-to-micron scale), and compositions. Many NPs and MPs comprise commodity plastics, such as polystyrene (PS), polyethylene (PE), polyethylene terephthalate (PET), polyamide, and polypropylene (PP) [18], but polymer formulations likely include other components including additives and plasticizers. The heterogeneity of these small-scale plastics has called for a consensus on accepted definitions of MPs and NPs [19].

Key uncertainties exist about the potential effects on human health from the consumption and exposure to NPs and MPs. For example, studies have shown the presence of MPs in human stool [20] and lung tissue [21], which raise concerns about the influence of these materials on human organs. In particular, the effects of plastics <1 µm on biological systems is critical to understand [22], given the propensity of nanomaterials to enter cells [23,24,25] and tissues [26]. To date, the majority of studies of the interactions between biological systems and NPs have focused on PS nanoparticles [27,28,29,30,31]. For example, numerous in vitro studies have investigated the effects of PS nanoparticles on cell models, showing that cellular responses depend on multiple parameters including the cell type, as well as the size and surface functionalization of the PS nanoparticles [31,32,33,34,35]. The NPs that reside in the environment and in consumer products, however, consist of a greater compositional diversity than pristine PS. Recently, studies have started to focus on the biological responses to other nanoparticles made of commodity plastics, including PET [36,37] and PP [38]. Nanoplastics with well-characterized properties can greatly support steps towards understanding how the individual compositions and physicochemical properties of NPs can affect complex biological systems. 

One commodity thermoplastic, nylon, is used across various market sectors including automotive applications, fabrics, packaging, and electronics with an estimated global market of 8.9 million tons in 2020 [39]. The term nylon encompasses a variety of polyamides typically synthesized via ring-opening polymerization or condensation reactions to produce different chemical arrangements [40]. Nylon-6 (Appendix A, ESI†) is one such composition that continues to hold a prominent role in numerous applications for fabrics, packaging materials and various consumer goods [41]. Likewise, nylon-11 is a semicrystalline ferroelectric polyamide [42] utilized for a wide range of products in the automotive, medical, and energy industries. In the context of MPs and NPs, polyamide MPs have been reported in bottled water [43], table salt [11,44], bivalves [45,46], and mollusks [47,48] intended for human consumption. The preparation of nylon particles is not widely reported in the literature with the exception of a few key manuscripts. Crespy and Landfester described the preparation of nylon-6 nanoparticles via the anionic polymerization of ε-caprolactam within a solvent through a miniemulsion approach [49]. The same authors also described the preparation of nylon-6 nanoparticles stabilized by polyvinyl alcohol (PVA) using a method that combined miniemulsion and solvent displacement [50]. A report by Ma et al. described the preparation of piezoelectric nylon-11 nanoparticles using an anti-solvent method for use in promoting the osteogenic differentiation of stem cells [51]. To evaluate the cellular uptake, the authors fluorescently labeled these nylon-11 nanoparticles via surface coating with oxidized polydopamine. In another report, nylon-6,6 nanoparticles with a cauliflower-type structure were produced by the magnetron sputtering of nylon in a gas aggregation source [52].

Herein, we report approaches to readily fabricate nylon-11 and nylon-6 particles for studies in mammalian systems. We detail the preparation and characterization of the particles and further describe the exposure of mammalian cells to these particles. This manuscript supports the critical need of highly characterized particles of commodity plastics to examine the downstream effects in biological systems.

## 2. Materials and Methods

### 2.1. Fabrication and Purification of NPs

#### 2.1.1. Fabrication of Nylon-11 NPs

Unless otherwise noted, all reagents for fabricating the nylon-6 and nylon-11 particles were purchased from Sigma-Aldrich (St. Louis, MO, USA). In a 40 mL scintillation vial containing a magnetic stir bar, a solution of nylon-11 was prepared by mixing 0.35 mg nylon-11 (Sigma-Aldrich, Cat. No. 181153) with 20 mL hexafluoroisopropanol (HFIP). Nylon-11 particles were prepared via the dropwise addition of nylon-11 solution (10 mL, 1 mL/min) to ultrapure deionized water (75 mL, 18.2 MΩ·cm resistivity) using a syringe pump (Model # NE-300, New Era Pump Systems, Inc., Farmingdale, NY, USA). Residual HFIP was removed by distillation by subjecting the precipitate to rotary evaporation under vacuum at 60 °C. Once the volume was reduced to ~30 mL, an additional volume (~75 mL) of ultrapure deionized water was added and rotary evaporation was continued. This process was repeated a total of five times. 

Particles containing Nile Red (NR) or Acryloxyethyl Thiocarbamoyl Rhodamine B (ATRB) were formulated using a similar approach as specified above. The stock solutions of both the tracers in HFIP (1 mg/mL) were prepared. An aliquot (1 mL) was then added to the nylon-11 solution before initiating the precipitation protocol specified above. 

#### 2.1.2. Fabrication of Nylon-6 NPs

In a 20 mL scintillation vial, a solution of nylon-6 (Sigma-Aldrich, Cat. No. 181110) was prepared by heating 15 mg of nylon-6 pellets in 1 mL formic acid at 70 °C for 30 min and then cooled to room temperature. In a separate 20 mL scintillation vial, a solution containing 24 mg polyvinyl alcohol (PVA, MW ~25 kDa, 88 mol% hydrolyzed, Cat. No. 02975, Polysciences, Inc., Warrington, PA, USA), 7 mL ultrapure deionized water (18.2 MΩ·cm resistivity), and 3 mL methanol was ultrasonicated for 30 s at 70% amplitude (Model No. CV17, Sonics & Materials, Inc., Newtown, CT, USA), while the vial was cooled by ice. The nylon-6 solution was added dropwise over 1 min while continuing sonication, and sonicated for an additional 30 s to yield nylon-6 particles. 

The fabricated particle suspension was centrifuged and resuspended to remove the excess formic acid and methanol. For each wash step, the suspension was centrifuged at 16,000 rcf for 5 min at room temperature, supernatant was removed, and the pellet was resuspended in an equal volume of 0.5 mg/mL PVA. The resuspension of particles was achieved by a 30 s vortex step followed by discrete sonication in a cup horn sonicator (Ultrasonic Liquid Processor S-400, Misonic Inc., Farmingdale, NY, USA) delivering a total of 1680 J/mL. The particles were washed two times and then resuspended one final time.

A similar procedure was adopted to fabricate NR- and ATRB-tagged nylon-6 particles. A solution of NR in formic acid (0.1 mg/mL) was prepared from a stock solution of 1 mg/mL. An aliquot of the 0.1 mg/mL Nile Red (1 mL) was mixed with the nylon-6/formic acid solution before it was added dropwise to the sonicated PVA mixture. Likewise, an aliquot (1 mL) of ATRB in formic acid (1 mg/mL) was mixed with the nylon-6/formic acid solution. 

### 2.2. Characterization of Particles

#### 2.2.1. Quantification of Fluorescence of NPs

An aliquot (1 mL) of nylon-6 or nylon-11 NPs was transferred to a tared Eppendorf tube and placed in a vacuum oven (Model AccuTemp-09 k, Across International, Livingston, NJ, USA) overnight, and was weighed the next day. The dried particles were dissolved in formic acid (1 mL, nylon-6) or HFIP (1 mL, nylon-11) and their fluorescence was determined using Synergy MX multi-mode plate reader (BioTek Instruments, Inc., Winooski, VT, USA). Calibration curves of NR and ATRB in formic acid were obtained via serial dilutions of the fluorophore (NR in formic acid: 10 µg/mL stock solution, *λ*_ex_ = 590 nm, *λ*_em_ = 670 nm; ATRB in formic acid: 10 µg/mL stock solution, *λ*_ex_ = 560 nm, *λ*_em_ = 590 nm; NR in HFIP: 5 µg/mL stock solution, *λ*_ex_ = 610 nm, *λ*_em_ = 670 nm; ATRB in HFIP: 62.5 ng/mL stock solution, *λ*_ex_ = 550 nm, *λ*_em_ = 580 nm). 

#### 2.2.2. Fluorescence Leaching

An aliquot (400 µL) of fluorophore-loaded nylon NPs was added to tared Amicon^®^ Ultra centrifugal filter unit (regenerated cellulose, 100 K, MilliporeSigma, Burlington, MA, USA) and was centrifuged at 16,000 rcf for 10 min. The filtrate collected from this spin was subjected to an additional centrifugation step for 10 min at 16,000 rcf using tared Amicon^®^ Ultra centrifugal filter unit (regenerated cellulose, 3 K). Samples were collected in duplicate and at time 0, 7 days, and 30 days. The NPs were stored in the refrigerator at 4 °C over the 30-day time period. The fluorescence of the second filtrate was determined using Synergy MX multi-mode plate reader. The calibration curves of NR and ATRB in ultrapure water were obtained by serial dilutions of the fluorophores (NR: 2.5 µg/mL stock solution, *λ*_ex_ = 590 nm, *λ*_em_ = 667 nm; ATRB: 1.25 µg/mL stock solution, *λ*_ex_ = 560 nm, *λ*_em_ = 590 nm). 

#### 2.2.3. Fourier-Transform Infrared Spectroscopy (FT-IR)

Samples of particles were analyzed using a Nicolet 6700 FTIR instrument equipped with a Smart Orbit™ single bounce diamond crystal ATR accessory, a deuterated triglycine sulfate (DTGS) detector, and a potassium bromide (KBr) beam splitter. The method involved 32 scans over a region of 4000–400 cm^−1^ and a resolution of 4. Prior to running each sample, a background was collected on the cleaned crystal, and the sample was introduced on the diamond crystal. Pressure was applied and the sample data were collected. The suspension of nylon-6 particles that were used for FT-IR were washed using water instead of 0.5 mg/mL PVA.

#### 2.2.4. Dynamic Light Scattering (DLS) and Zeta Potential

Zetasizer Nano ZS (Malvern Instruments, Malvern, UK) equipped with a He-Ne laser (633 nm) was used to acquire DLS measurements (non-invasive backscatter method with a scattering angle of 173°). The hydrodynamic diameters (D_H_), polydispersity indices (PDI), and zeta potential of polymer particles were calculated by the instrument software (Zetasizer DTS). For these measurements, nylon-6 particles were suspended in 0.5 mg/mL PVA or cell media, and nylon-11 particles were suspended in ultrapure deionized (DI) water or cell media. 

#### 2.2.5. Formic Acid Protocol

The concentration of formic acid in purified nylon-6 NPs was determined using the formic acid assay kit (Megazyme, Wicklow, Ireland). Briefly, the supplied reagents were suspended in ultrapure water at the specified concentrations in the assay protocol. The blank samples comprised ultrapure water, solution 1 (buffer), and solution 2 (NAD+) mixed at specified ratios and mixed thoroughly. In a 96-well plate, sodium formate standard and nylon-6 NPs were added to the blank solution. The absorbance of the solutions (A1) at 340 nm was measured after approximately 5 min using the Synergy MX multi-mode plate reader. The absorbance was measured every 2–3 min after a 12 min incubation with the supplied enzyme, formate dehydrogenase. The absorbance values (A2) were recorded once they reached a plateau. Formic acid concentration was calculated using the equation:$$C_{formic\;acid}=\frac{\Delta \;Absorbance_{sample}}{\Delta \;Absorbance_{standard}}\;\times \;C_{standard}\;$$
where $C_{formic\;acid}$ is the final concentration of formic acid in the sample, $\Delta Absorbance_{sample}\;\mathrm{is}\;A2-A1\;\mathrm{for}\;\mathrm{the}\;\mathrm{nylon}-6\;\mathrm{NP}\;\mathrm{samples}$, $\Delta Absorbance_{standard}\;\mathrm{is}\;A2-A1\;\mathrm{for}\;\mathrm{the}\;\mathrm{sodium}\;\mathrm{formate}\;\mathrm{standard}$, and $C_{standard}$ is the concentration of the sodium formate standard

#### 2.2.6. Field Emission Scanning Electron Microscopy (FE-SEM)

Samples of particles were characterized using FE-SEM (Auriga FIB/FESEM, Carl Zeiss Microscopy, Peabody, MA, USA). Before imaging, the samples were coated with a few nanometers of AuPd using a Leica ACE200 Sputter Coater (Leica Microsystems, Buffalo Grove, IL, USA).

### 2.3. Studies with Mammalian Cells 

#### 2.3.1. In Vitro Sedimentation, Diffusion, and Dosimetry (ISDD) Modeling

The effective dose for the particles was calculated following a protocol by DeLoid et al. [53]. Briefly, the effective density for each particle formulation was determined in growth media using the volumetric centrifugation method. The particle pellet volume was measured using TPP packed cell volume (PCV) tubes and an easy-read measuring device for PCV tubes (TPP Techno Plastic Products, Midwest Scientific, Valley Park, MO, USA). A 25 Cannon-Fenske tube viscometer (Sigma-Aldrich, St. Louis, MO, USA) was used to measure the dynamic viscosity of cell culture media. After calculating the effective density for each of the four particles in growth media, the effective dosimetry was determined by computational modeling using the volumetric centrifugation method—in vitro sedimentation, diffusion, and dosimetry (VCM–ISDD) or distorted grid (DG) model [53].

#### 2.3.2. Cell Culture

The particles were tested on mouse alveolar macrophage cells, RAW 264.7 (ATCC^®^ TIB-71™, ATCC, Manassas, VA, USA). The cell culture media comprised Dulbecco’s modified Eagle’s medium (Gibco, Life Technologies, Grand Island, NY, USA), supplemented with 10% fetal bovine serum (FBS) (Gibco, Life Technologies, Grand Island, NY, USA) and 100 U penicillin/streptomycin (P/S) (Gibco, Life Technologies, Grand Island, NY, USA). Cells were maintained at a concentration of 1 × 10^4^ cell/mL at 37 °C in 5% humidified CO_2_. Cells were passaged twice a week by washing with pre-warmed phosphate-buffered saline (PBS) (Gibco, Life Technologies, Grand Island, NY, USA). For these studies, the cells were used between passage numbers 4–8.

#### 2.3.3. Cytotoxicity Assays

RAW 264.7 cells were seeded at a concentration of 1 × 10^5^ cells/mL within a 96-well plate and then incubated for 24 h. In addition to nylon NPs, two types of commercially sourced PS nanoparticles (500 nm and 50 nm PS Yellow nanoparticles; Spherotech, Lake Forest, IL, USA) with size distributions similar to the nylon NPs were also evaluated in the cytotoxicity assays. Cells were exposed to particles suspended in fresh media in two-fold dilutions ranging from 0.001 to 1.0 mg/mL. The medium was collected after 24 h of particle exposure and lactate dehydrogenase (LDH, TOX7, Sigma-Aldrich, St. Louis, MO, USA) release measurements were performed (according to the protocol specified by the manufacturer. All studies were conducted in at least biological duplicates and experimental triplicates with data expressed as a percentage of their corresponding controls. A control LDH assay was performed in the absence of cells to assess any interference from the particles with the assay. For 1.0 mg/mL 500 nm PS Yellow nanoparticles, ~30% increase in the background value was observed, while no increase was observed for unlabeled nylon-11, unlabeled nylon-6, and 50 nm PS nanoparticles.

#### 2.3.4. Fluorescence Microscopy

In glass bottom Petri dishes (MatTek, Ashland, MA, USA), 1 × 10^5^ cells/mL cells were seeded for 24 h. The cells were then exposed to particle concentrations of 0.01 and 0.1 mg/mL for 16 h. For these microscopy studies, nylon-6 ATRB contained ~2.8 µg ATRB/mg particles, nylon-6 NR contained ~6.9 µg NR/mg particles, nylon-11 ATRB contained ~4.5 µg ATRB/mg particles, and nylon-11 NR contained ~14.1 µg NR/mg particles. The vehicle controls included in this study were 2.2 mg/mL PVA solution (control for nylon-6 NPs) and ultrapure deionized water (control for nylon-11 NPs). Cells were subsequentially fixed with 4% paraformaldehyde at room temperature for 30 min. The cells were washed thrice with PBS and then stained with 1:1000 DAPI (Life Technologies, Grand Island, NY, USA) for 20 min at room temperature. Prior to bright-field and fluorescence imaging with a 40× objective, the cells were washed an additional three times with PBS. Imaging was conducted using a Zeiss Observer Z1 3D inverted microscope with three channels.

#### 2.3.5. Data Analysis

Data are expressed as mean ± standard deviation. To test for significant differences in LDH release due to particle exposure and to understand the role of plastic composition and concentration, a repeated measure two-way ANOVA test was conducted with Tukey’s multiple comparisons test. The statistical analyses were performed using GraphPad Prism 7.04 (GraphPad Software, San Diego, CA, USA).

## 3. Results and Discussion

### 3.1. Preparation and Characterization of Nylon Particle Formulations

#### 3.1.1. Fabrication and Characterization of Nylon-11 NPs

Nylon-11 particles were fabricated via a precipitation method similar to a previously published procedure [36]. Attempts at removing residual HFIP through centrifugation or dialysis were unsuccessful as a result of the low pelletization of the small-sized particles (Table 1) and the formation of aggregates. Since these methods of particle purification were ineffective for removing residual HFIP, multiple cycles of rotary evaporation were used to substantially reduce the fluorine signal via ^19^F-NMR (Appendix A, ESI†). 

An attempt to incorporate RhB into nylon-11 particles to trace NPs within cells resulted in negligible loading of fluorophore. Therefore, alternative fluorescent tracers NR and ATRB were incorporated into the NPs during fabrication. Based on fluorescence spectroscopy, the concentrations of fluorophore in the particle formulations were ~4.5 µg ATRB/mg particles and ~14.1 µg NR/mg particles. Although rotary evaporation effectively eliminated HFIP while retaining the size distributions of the NPs, this purification approach is not suitable for eliminating the potential presence of excess fluorophore in the solution.

The nylon-11 NPs showed a spherical morphology via SEM (Figure 1) and no morphological differences were apparent for the nylon-11 particles with the ATRB and NR tracers. Hydrodynamic diameters of the purified particles comprising nylon-11 (127 ± 51 nm), nylon-11 ATRB (142 ± 43 nm), and nylon-11 NR (137 ± 39 nm) were unchanged after multiple cycles of rotary evaporation (Appendix A, ESI†). The size distributions for the unlabeled, nylon-11 ATRB, and nylon-11 NR particles were also calculated from the SEM images to be 55 ± 19 nm, 88 ± 22 nm, and 84 ± 12 nm, respectively. The slight differences between the size distributions obtained from DLS and SEM could be attributed to the use of dried samples to acquire SEM images. The high zeta potential (Table 1) across all three sets of nylon-11 particles ensured colloidal stability and eliminated the need for the incorporation of additional stabilizers [54]. As the particles were not exposed to particle stabilizers during the fabrication or purification processes, the surface charges of the particles pre- and post-purification were expectedly unchanged (Table 1). 

#### 3.1.2. Fabrication and Characterization of Nylon-6 NPs

A precipitation protocol was initially used to synthesize nylon-6 particles, but large visible aggregates of polymer formed during the precipitation step. The higher hygroscopicity of nylon-6, as compared to nylon-11 [55,56,57], could affect the precipitation step that relies on the self-assembly of the polymers through hydrophobic interactions. Therefore, to successfully prepare nylon-6 particles, a combination of ultrasonication and washing was implemented, similar to a previously published protocol [50]. The formation of particles occurs after the slow addition of a nylon-6/formic acid solution to a second solution containing PVA while exposed to ultrasonicating forces. After the completion of fabrication steps, nylon-6 particles were washed with 0.5 mg/mL PVA to remove residual organic solvents, to stabilize particles and to prepare the NPs for use in biological assays. After washing, the particles revealed low levels of formic acid (Appendix A, ESI†), well below the cytotoxic threshold (~7.56–18.66 µmol/mL) [58,59]. Interestingly, the washing steps resulted in a reduction in the average diameter and improved PDI (Table 1). The decrease in the diameter of the nylon-6 NPs after the washing steps (Appendix A, ESI†) may be attributed to the purification process as previously reported for other nanoparticle formulations [60]. The average diameter of 465 ± 132 nm after the final wash was also confirmed with SEM (Figure 2). The SEM images also show that the surface morphology of nylon-6 NPs is irregular with the appearance of multiple nucleation sites. After the wash steps, the zeta potential increased from 5.09 ± 0.62 mV to 22.63 ± 0.05 mV, which could be attributed to the change in PVA concentration (2.2 mg/mL during fabrication; 0.5 mg/mL wash solution) [61]. 

This method of fabricating nylon-6 NPs enables the incorporation of fluorescent tracers into the particles by first dissolving the fluorophore within the polymer solution prior to the ultrasonication step. NR, ATRB and Texas Red (TR) maintain stable fluorescence at acidic pH, and are therefore compatible with the use of formic acid during the fabrication steps [62,63,64]. Similar trends in size and surface charge were observed for the fluorophore-tagged nylon-6 particles as compared to the unlabeled NPs (Table 1). 

To support microscopy studies in cell cultures, the fabrication of the fluorescent nylon-6 NPs was optimized by testing different loading concentrations of the fluorophore (Appendix A, ESI†). For the ATRB tracer, the use of 1 wt% of fluorophore during fabrication was required to achieve a detectable fluorescence signal (~2.8 µg ATRB/mg particles). Using lower concentrations of ATRB during fabrication (i.e., 0.1 wt%) resulted in the undetectable fluorescence in the nylon-6 NPs. For the NR tracer, the use of 0.1 wt% of NR during fabrication was prioritized for cell exposure studies, which afforded ~6.9 µg NR/mg particles. The use of higher concentrations of NR in the fabrication (i.e., 1 wt% NR) caused aggregated particles despite rigorous sonication and vortexing. For the TR tracer, the use of 0.1 wt% of fluorophore resulted in ~1.65 µg TR/mg particles; however, negligible fluorescence was observed in cells (Appendix A, ESI†). While TR is highly hydrophobic, the sulfonyl chloride moiety of unreacted TR molecules is susceptible to hydrolysis and converts into water-soluble sulfonate [65]. Since the particles are suspended in aqueous media over several days prior to cell imaging, it is possible that TR hydrolyzed to sulfonate during storage. Coupled with the low fluorophore loading, the extensive washing steps to remove unincorporated fluorophore could have removed the more hydrophilic TR sulfonate and hence, it might not have been detected during fluorescence imaging. 

#### 3.1.3. Fluorescence Leaching 

Ensuring the stability of fluorescent-labeled NPs is an important consideration when conducting cell exposure and uptake studies. If the NPs undergo fluorophore leakage, the observed fluorescence in cells could be erroneously represented as the uptake and accumulation of NPs instead of free fluorophore [66,67,68]. The presence of residual fluorophore can also induce cytotoxicity and without the appropriate controls, the ability to differentiate toxicity as a function of particle or fluorophore uptake can be challenging [69,70]. Fluorophore desorption could additionally lead to altered particle characteristics (i.e., changes in dispersibility, size, surface charge, and morphology) [71,72]. The potential leaching of fluorophore across all nylon NPs was assessed by measuring the fluorescence of the solution after the sequential filtration of the NPs through 100 K and 3 K centrifugal filters over 30 days (Appendix A, ESI†). The low fluorescence detected in the solution after removing the NPs indicates the fluorophore stability within the NPs and suggests that there is negligible fluorophore leaching from the NPs. 

#### 3.1.4. FT-IR Characterization of Nylon-11 and Nylon-6 NPs

The composition of all nylon NPs was evaluated using FT-IR (Figure 3). For nylon-6 NPs, some characteristic absorption bands include 3299 cm^−1^ (N-H stretching), 2940 cm^−1^, and 2868 cm^−1^ (C–H stretch from ethylene groups), 1639 cm^−1^ (amide I), and 1544 cm^−1^ (amide II) [73]. The nylon-6 particles labeled with NR and ATRB showed similar absorption bands distinctive of the polymer; however, the bands associated with the fluorescent molecule were absent (Appendix A, ESI†). For nylon-11 NPs, the characteristic absorption bands include 3301 cm^−1^ and 2919 cm^−1^ (N–H stretch), 2851 cm^−1^ (C–H vibration of the methylene groups), 1638 cm^−1^ (amide I), 1546 cm^−1^ (amide II), and 1467 cm^−1^ (C–H bending) [74,75]. The incorporation of NR and ATRB within the nylon-11 NPs did not result in detectable absorption bands from the fluorophore (Appendix A, ESI†). Because the presence of fluorescent moieties for all nylon NPs was confirmed with a fluorescence assay, the absence of FT-IR bands was likely a result of the low concentration of fluorophore in the particles. 

### 3.2. Exposure of Mammalian Cells to Nylon NPs 

#### 3.2.1. Particle Characterization and Stability

Prior to initiating cell exposure studies, the size and morphology of NPs in cell media were tested over a fixed time period. Since PS particles are widely used for studying the effects of plastics, two sets of commercially sourced PS particles with size distributions similar to nylon NPs were also evaluated. Although the size distributions between the PS and nylon particles slightly differed, the comparison can still provide useful insight into the differences in composition and size. The introduction of all particle types to cell culture media yielded particles with increased overall diameters, likely attributed to the formation of the protein corona or slight aggregation in cell media (Table 2) [76]. After a 24 h incubation in cell media, minimal changes in the diameters were observed, indicative of colloidal stability over this time. A wide range of zeta potential values was observed across the different types of particles suspended in water: −69 mV to 48 mV. The zeta potential for particles in cell media at both at 0 and 24 h changed significantly, as compared to the NPs in water, suggesting that the cell media components (e.g., proteins) shielded the original particle surface charge (Table 2) [77,78]. 

#### 3.2.2. In Vitro Sedimentation, Diffusion, and Dosimetry (ISDD) Modeling

To further elucidate the impact of particle size, density, and polydispersity in cell media, the effective dosimetry concentration dose for nylon and commercially available PS particles were assessed using ISDD modeling. This model was used to calculate the effective dose for the particles over the time course of 24 h and showed that the concentration and fraction of particles in the growth media immediately above the RAW 264.7 cells (expressed as the volume between the cell monolayer and 10 µm above) varied by orders of magnitudes between the different particle formulations (Appendix A, ESI†). 

Both the smaller particle formulations (50 nm PS and nylon-11 NPs, Figure 4A,B) had comparable settling profiles with mean effective dose concentrations of 3.84 mg/mL and 7.50 mg/mL, respectively. These smaller sized particles measured similar effective densities in cell media (50 nm PS NP: 1.04 ± 0.0 g/cm^3^ and nylon-11 NP: 1.03 ± 0.0015 g/cm^3^) and likely experienced greater Brownian motion and slower rates of settling. For example, only 16.8% (50 nm PS) and 36.7% (nylon-11) particles accumulated at the bottom of the wells over 24 h. 

Unlike the smaller particles (nylon-11, 50 nm PS), there was a significant difference observed in the settling rates for these two larger sized particle formulations (nylon-6, 500 nm PS). The 500 nm PS particles with a measured effective density of 1.04 ± 0.0013 g/cm^3^ exhibited the lowest mean concentration (1.12 mg/mL) along with the lowest fraction deposited in cells (7.2%, Figure 4C). Conversely and expectedly, nylon-6 NPs due to its higher measured effective density (1.46 ± 0.18 g/cm^3^) rapidly reached a plateau within 4 h and had the highest calculated mean concentration (18.4 mg/mL) with 72.3% sedimented at the bottom of the wells (Figure 4D). These data underscore the importance of considering both the size and chemical composition when evaluating the effects of microplastics and nanoplastics on biological systems. The use of commercial PS particles to universally evaluate the impact of NPs on biological systems fails to represent the breadth of plastics and supports the need to develop materials that are more representative of ubiquitous commodity plastics. 

#### 3.2.3. Cytotoxicity Studies with NPs

Murine alveolar macrophages, RAW 264.7, were used to assess how different concentrations of NPs affect cell membrane integrity by using the LDH cytotoxicity assay (Figure 5). For PS nanoparticles (50 nm and 500 nm), concentrations of up to 0.25 mg/mL did not result in any significant cytotoxic effects. This finding aligns with a report from Florance et al. which showed that the viability of RAW 264.7 macrophages was unaffected when dosed with PS nanoparticles (208.63 ± 6.494 nm) at 0.2 mg/mL for 24 h [79]. Despite minimal cytotoxic effects, the authors interestingly showed that PS nanoparticles (0.1 mg/mL) affected cellular homeostatis by increasing the generation of reactive oxygen species (ROS) four hours post-dosing. Another study showed that PS nanoparticles (42 nm) dosed between 0.1 and 10 µg/mL to RAW 264.7 cells triggered ROS generation and proinflammatory cytokines [80]. In the current study, both sizes of the PS nanoparticles (50 nm and 500 nm) exhibited cytotoxicity at a threshold dose of 0.5 mg/mL of NPs, suggesting that alterations in cellular homeostatis may occur before the loss of cell membrane integrity. As indicated in the Materials and Methods section, control studies revealed that 500 nm PS nanoparticles at a concentration of 1.0 mg/mL interfered with the LDH assay, which may explain the higher level of apparent cytotoxicity for this particle type; nylon-6, nylon-11 NPs and 50 nm PS nanoparticles did not interfere with the LDH assay. Figure 5 also shows that nylon-6 and nylon-11 NPs did not exhibit cytotoxicity until a dose of 1 mg/mL. These overall findings, in addition to statistical analyses, suggest that both particle concentration (F_11,66_ = 83.4), *p* < 0.0001 and particle composition (F_3,18_ = 8.64), *p* = 0.0009, along with interaction between the two parameters (F_33,198_ = 15.5), *p* < 0.0001, were contributing sources to the measured outcome. To the best of the author’s knowledge, no studies have evaluated the effects of nylon-based NPs on RAW 264.7 cells. However, using dental pulp stem cells, Ma et al. showed that nylon-11 nanoparticles (approximately 50 nm) were cytocompatible at a dose of 400 µg/mL [51]. Nanoparticles comprising another commodity plastic, PET, were recently tested in multiple studies with RAW 264.7 cells [36,37,81]. At concentrations of 15 µg/mL, Aguilar-Guzmán and colleagues showed the PET nanoparticles affected RAW 264.7 cells by increasing the production of ROS, altering cell proliferation, and upregulating certain genes likely related to foreign particle responses and cell maintenance [81]. For nylon-based NPs, future studies are required to understand the potential effects of these particles on gene expression.

#### 3.2.4. Fluorescence Microscopy of Macrophages Exposed to NPs 

The uptake of fluorescently labeled nylon-6 and nylon-11 NPs by macrophages (RAW 264.7 cells) is shown in the overlay of bright-field and fluorescence microscopy images (Figure 6). After exposing cells to nylon-11 and nylon-6 NPs for 16 h, the spatial distribution in the cellular cytoplasm was evident. The images in Figure 6 show a demarcation between the cell nucleus (blue; DAPI stain) and the presence of fluorescent clusters of ATRB- and NR-tagged nylon NPs in the cytoplasm. The presence of nanoparticles within the cytoplasm of mammalian cells was also reported for nanoparticles comprising other commodity polymers, such as PS [23], PVC [82], and PET [36]. As mentioned previously, TR-tagged nylon NPs did not exhibit any fluorescence across the tested particle concentrations (Appendix A, ESI†). The uptake of nylon NPs in macrophages occurred in a dose-responsive manner, with negligible cellular uptake occurring after exposure to the lowest concentration 0.01 mg/mL of NPs and the highest particle uptake observed at 1 mg/mL of NPs. In agreement with the membrane integrity results in Figure 5, the microscopy images overall showed an intact cellular morphology after the exposure of NPs at lower concentrations (0.01 mg/mL and 0.1 mg/mL), but signs of membrane damage and instability are apparent at the highest 1.0 mg/mL concentration.

## 4. Conclusions

The existence of MPs and NPs in the environment, food, and beverages has raised key questions around the potential for downstream effects in human health. Given the diversity of polymer formulations in modern society, there is a need for well-characterized NPs comprising commodity polymers to systematically test the **e**ffects on complex biological systems. This manuscript describes the preparation of well-characterized nylon-6 NPs and nylon-11 NPs with hydrodynamic diameters of 465 ± 132 nm and 127 ± 51 nm, respectively, for unlabeled particles after purification. To facilitate studies in biological systems, NPs were also successfully labeled with NR or ATRB fluorescent tracer to aid with the in vitro visualization and intracellular tracking. A 30-day shelf stability study of nylon NPs in aqueous media showed no leaching of fluorescent tracer from the particles. The exposure of RAW 264.7 macrophages to nylon NPs over the concentration range of 0.001–1.0 mg/mL resulted in cytotoxicity at 1 mg/mL. Fluorescence microscopy images showed the uptake of NPs within the macrophages and indications of membrane damage at the highest 1.0 mg/mL concentration of nylon-6 and nylon-11 NPs. These well-characterized nylon NPs support future steps to fully understand the influence of these small-scale plastic materials on biological systems and ultimately human health.

## Figures and Tables

**Figure 1 nanomaterials-12-02699-f001:**
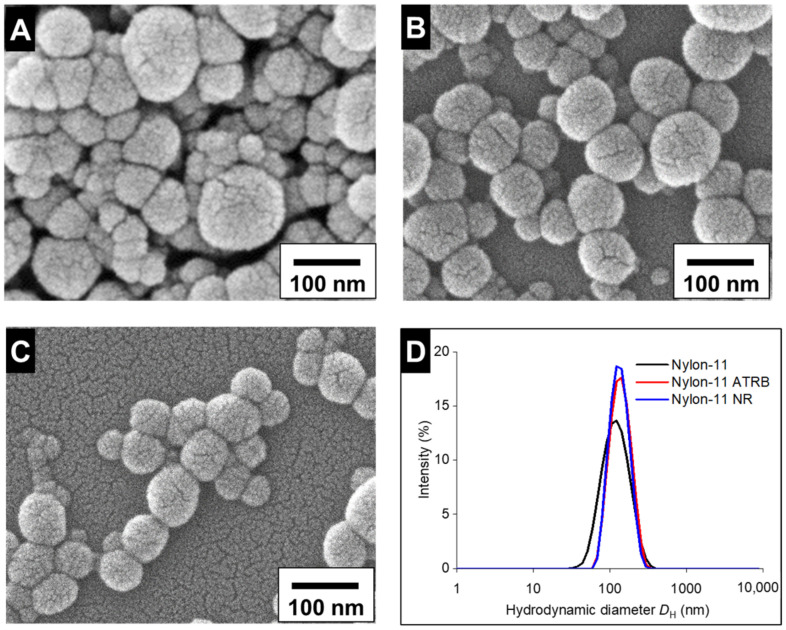
The SEM images for nylon-11 (**A**), nylon-11 ATRB (**B**), and nylon-11 NR (**C**) along with their corresponding DLS profiles (**D**). The DLS profile is an average of 3 measurements.

**Figure 2 nanomaterials-12-02699-f002:**
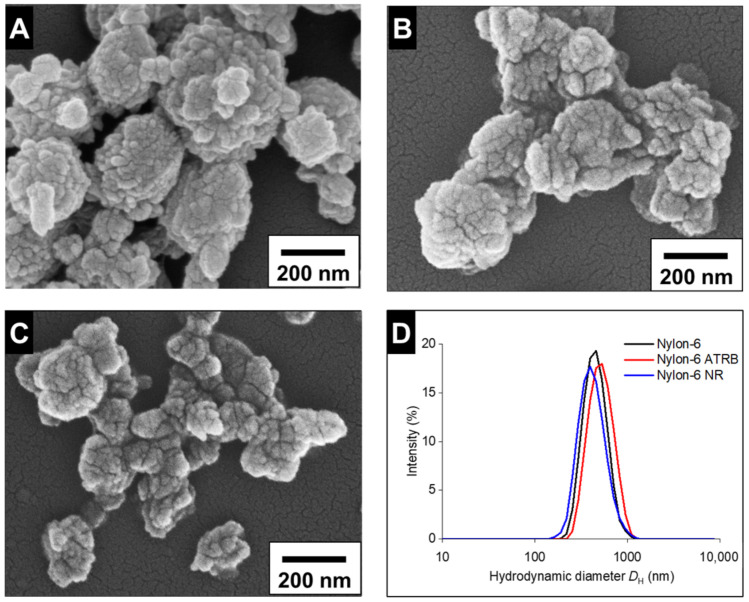
The SEM images for nylon-6 (**A**), nylon-6 ATRB (**B**), and nylon-6 NR (**C**) NPs along with their corresponding DLS profiles (**D**). The DLS profile is an average of 3 measurements.

**Figure 3 nanomaterials-12-02699-f003:**
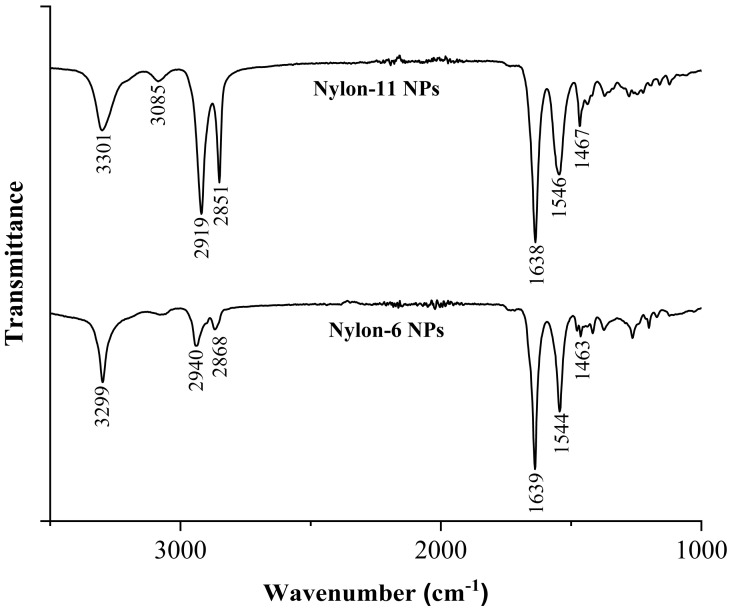
FT−IR spectra of nylon−11 NPs (**top**) and nylon−6 NPs (**bottom**).

**Figure 4 nanomaterials-12-02699-f004:**
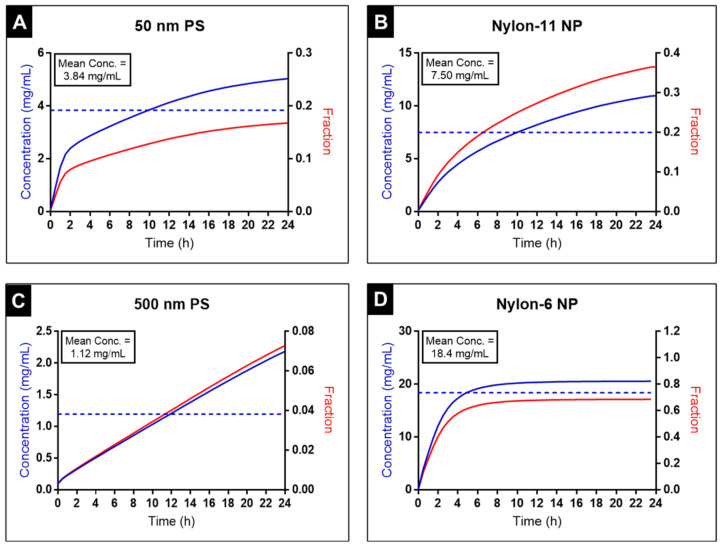
Effective dosimetry concentration (blue line) and fraction (red line) calculated for each of the four particles at the bottom of the well proximal to the RAW 264.7 cells using the ISDD model. The mean particle concentration throughout the 24 h exposure is indicated with a blue dotted line for each particle type, and the calculated mean concentration is listed in the box on each graph (**A**–**D**).

**Figure 5 nanomaterials-12-02699-f005:**
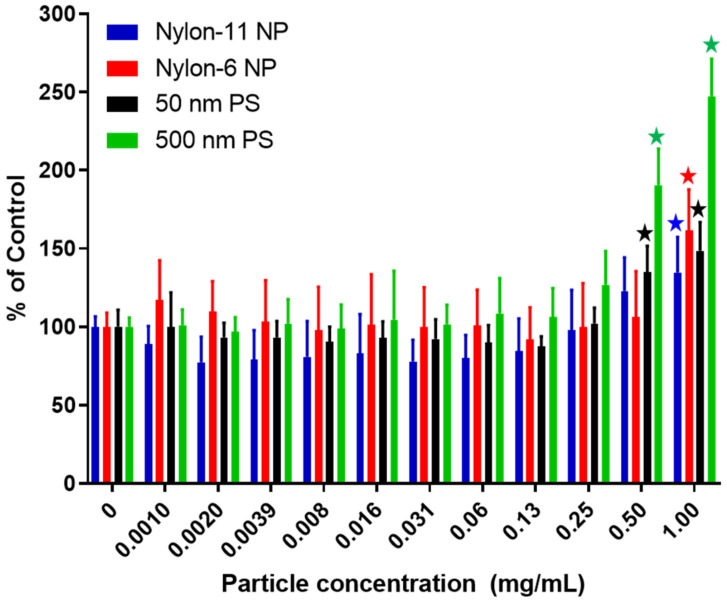
Cytotoxicity of 50 nm PS nanoparticles (black), 500 nm PS nanoparticles (green), nylon-11 NPs (blue), and nylon-6 NPs (red) after exposure to different doses of NPs for 24 h. The graphs show mean ± standard deviation. Significant difference between corresponding vehicle control and dose treatment is shown in stars for each particle. Two-way ANOVA showed that nanoparticle composition contributed to the variation.

**Figure 6 nanomaterials-12-02699-f006:**
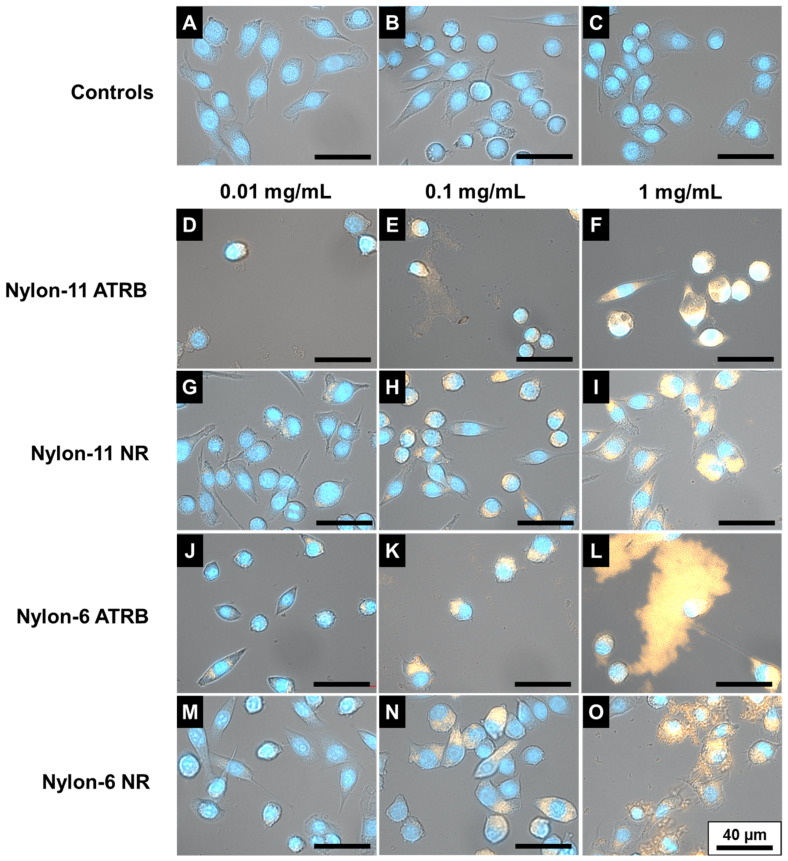
Overlay of bright-field and fluorescence microscopy images of RAW 264.7 cells exposed to the media control only (**A**); PVA vehicle control (**B**); water vehicle control (**C**); nylon-11 ATRB NPs at 0.01 mg/mL (**D**), 0.1 mg/mL (**E**), and 1 mg/mL (**F**); nylon-11 NR NPs at 0.01 mg/mL (**G**), 0.1 mg/mL (**H**), and 1 mg/mL (**I**); nylon-6 ATRB NPs at 0.01 m/mL (**J**), 0.1 mg/mL (**K**), and 1 mg/mL (**L**); and nylon-6 NR NPs at 0.01 mg/mL (**M**), 0.1 mg/mL (**N**), and 1 mg/mL (**O**). Cellular nuclei were stained with DAPI (blue) and the nylon NPs are shown in orange. Scale bar in each panel represents 40 µm.

**Table 1 nanomaterials-12-02699-t001:** Hydrodynamic diameter, PDI, and zeta potential * of nylon NP formulations.

	Before Purification			Post Purification		
	Average Hydrodynamic Diameter ± SD (nm)	Average PDI ± SD	Average Zeta Potential ± SD (mV)	Average Hydrodynamic Diameter ± SD (nm)	Average PDI ± SD	Average Zeta Potential ± SD (mV)
Nylon-11	136 ± 45	0.16 ± 0.03	35.23 ± 1.35	127 ± 51	0.19 ± 0.00	33.07 ± 1.41
Nylon-11-ATRB	160 ± 49	0.08 ± 0.01	28.67 ± 0.63	142 ± 43	0.08 ± 0.01	34.31 ± 0.85
Nylon-11-NR	164 ± 56	0.09 ± 0.01	30.23 ± 0.49	137 ± 39	0.08 ± 0.01	30.67 ± 0.38
Nylon-6	1686 ± 274	0.27 ± 0.11	5.09 ± 0.62	465 ± 132	0.10 ± 0.05	22.63 ± 0.05
Nylon-6-ATRB	755 ± 107	0.97 ± 0.04	7.39 ± 0.74	536 ± 160	0.08 ± 0.01	21.77 ± 0.26
Nylon-6-NR	2187 ± 261	0.41 ± 0.03	3.74 ± 0.38	436 ± 139	0.17 ± 0.05	22.73 ± 0.62
Nylon-6-TR	1398 ± 219	0.39 ± 0.10	5.71 ± 0.24	548 ± 248	0.23 ± 0.01	26.0 ± 0.78

* Zeta potential for nylon-11 NPs in ultrapure DI water and nylon-6 NPs in 0.5 mg/mL PVA.

**Table 2 nanomaterials-12-02699-t002:** Hydrodynamic diameter, PDI, and zeta potential in water and cell media for particles used in cell exposure studies.

	Deionized H_2_O			Cell Media		
	Average Hydrodynamic Diameter ± SD (nm)	Average PDI ± SD	Average Zeta Potential ± SD (mV)	Average Hydrodynamic Diameter ± SD (nm)	Average PDI ± SD	Average Zeta Potential ± SD (mV)
50 nm PS	61 ± 14	0.02 ± 0.02	−68.80 ± 1.88	0 h = 242 ± 105 24 h = 171 ± 82	0 h = 0.50 ± 0.08 24 h = 0.47 ± 0.04	0 h = −12.30 ± 1.10 24 h = −12.93 ± 1.08
500 nm PS	480 ± 97	0.04 ± 0.01	−59.17 ± 0.26	0 h = 597 ± 168 24 h = 557 ± 155	0 h = 0.13 ± 0.06 24 h = 0.06 ± 0.02	0 h = −6.65 ± 1.95 24 h = −11.7 ± 1.36
Nylon-11 NP	109 ± 41	0.21 ± 0.03	47.87 ± 0.58	0 h = 751 ± 277 24 h = 620 ± 193	0 h = 0.61 ± 0.09 24 h = 0.52 ± 0.08	0 h = −10.04 ± 0.85 24 h = −12.10 ± 0.43
Nylon-6 NP	476 ± 134	0.11 ± 0.03	21.93 ± 0.12	0 h = 602 ± 192 24 h = 569 ± 312	0 h = 0.28 ± 0.02 24 h = 0.50 ± 0.01	0 h = −5.17 ± 0.74 24 h = −6.82 ± 0.27

## Data Availability

The data presented in this study are available on request from the corresponding author. Certain data may be redacted or otherwise restricted for intellectual property reasons.

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
