# Peer review of "Fabrication of Nylon-6 and Nylon-11 Nanoplastics and Evaluation in Mammalian Cells"

_nanomaterials, 2022, doi:10.3390/nano12152699_

Round 1

Reviewer 1 Report

The work entitled “Fabrication of Nylon-6 and Nylon-11 Nanoplastics and Evaluation in Mammalian Cells” reports on the use of fluorescent tracers on Nylon NPs to aid tracking their uptake and cytotoxicity on macrophage cells. This is a very important issue that will definitely attract the interest of a broad audience.

The work Is very well organized and the subject is pertinent and well introduced. The scope of the project and the novelty of this investigation, with the demonstration of its impact, is clearly presented by the authors. The methodology is detailed and, thus, easily followed by other investigators. With the exception of small English mistakes, the discussion is well done, up to date and scientifically sound. The results are presented in a very clear manner, making the correlation of the data very easy. The only thing that would need to improve is the criticism/support of literature in the discussion, namely the mammalian cell testing. After this improvement, I believe the manuscript would be ready for publication.

Author Response

The authors are grateful for this positive and supportive feedback from Reviewer #1. As suggested, we have revised the cited literature describing the mammalian cell testing with nanoplastics. We included additional references to place our study in the context of currently published work.

We included additional references covering the effects of polystyrene nanoparticles on cell models (please see Introduction Section, lines 50-53):

  • Loos et al., "Functionalized polystyrene nanoparticles as a platform for studying bio–nano interactions," Beilstein Journal of Nanotechnology, vol. 5, pp. 2403-2412, 2014.
  • P. Walczak et al., "Translocation of differently sized and charged polystyrene nanoparticles in in vitro intestinal cell models of increasing complexity," Nanotoxicology, vol. 9, no. 4, pp. 453-461, 2015/05/19 2015, doi: 10.3109/17435390.2014.944599.
  • Anguissola, D. Garry, A. Salvati, P. J. O'Brien, and K. A. Dawson, "High Content Analysis Provides Mechanistic Insights on the Pathways of Toxicity Induced by Amine-Modified Polystyrene Nanoparticles," PLOS ONE, vol. 9, no. 9, p. e108025, 2014, doi: 10.1371/journal.pone.0108025.
  • Murali, K. Kenesei, Y. Li, K. Demeter, Z. Környei, and E. Madarász, "Uptake and bio-reactivity of polystyrene nanoparticles is affected by surface modifications, ageing and LPS adsorption: in vitro studies on neural tissue cells," Nanoscale, 10.1039/C4NR06849A vol. 7, no. 9, pp. 4199-4210, 2015, doi: 10.1039/C4NR06849A.

We included an additional reference pertaining to exposure of polystyrene nanoparticles to RAW 264.7 cells (Please see lines 468-469).

  • Hu, H. Wang, C. He, Y. Jin, and Z. Fu, "Polystyrene nanoparticles trigger the activation of p38 MAPK and apoptosis via inducing oxidative stress in zebrafish and macrophage cells," Environmental Pollution, vol. 269, p. 116075, 2021/01/15/ 2021.

We included more discussion of other commodity nanoplastics (please see lines 482-490 and 507-509) as well as additional references.

  • C. Aguilar-Guzmán et al., "Polyethylene terephthalate nanoparticles effect on RAW 264.7 macrophage cells," Microplastics and Nanoplastics, vol. 2, no. 1, p. 9, 2022/02/23 2022.
    • Mahadevan and S. Valiyaveettil, "Understanding the interactions of poly(methyl methacrylate) and poly(vinyl chloride) nanoparticles with BHK-21 cell line," Scientific Reports, vol. 11, no. 1, p. 2089, 2021/01/22 2021.

Reviewer 2 Report

This research present interesting results and is publishable. However, following comments should be addressed before publication: 

-Size distribution graphs should be presented. Are these average based on Volume? They should be based on Number or Intensity, 

-TEM should be presented.

-Figure 1D, the DLS data should be reported for Number% and Intensity. 

-Gene expression should be investigated for proper conclusion. 

-PS nanoparticles as control samples should have the same sizes of Nylon nanoparticles (based on TEM for both). 

-A size-dependent study for at least 4 different sizes of Nylon is recommended. 

-Cytotoxicity should be compared with biocompatible nanocarriers that could be found in following references: International journal of pharmaceutics, 2019, 569, 118580; New Journal of Chemistry, 2018, 42 (12), 9690-9701; Journal of Drug Delivery Science and Technology, 2020, 57, 101715; Advanced Powder Technology, 2020, 31 (9), 4064-4071; Materials Science and Engineering: C, 2020, 109, 110597; Mol. Syst. Des. Eng., 2022, : DOI: 10.1039/d2me00024e

Author Response

The authors thank Reviewer 2 for the enthusiasm regarding this manuscript. We have addressed each suggestion below and appreciated this feedback, which has improved this manuscript. 

-Size distribution graphs should be presented. Are these average based on Volume? They should be based on Number or Intensity,” 

  • Response from Authors: As requested, we have revised the size distribution graphs (Figure 1D, Figure 2D) to show Intensity vs. Hydrodynamic diameter.

-TEM should be presented.

  • Response from Authors: The authors greatly appreciate the use of multiple approaches to broadly characterize nanomaterials. The combination of SEM, DLS, Zeta-potential, and FT-IR, as described in the manuscript, is expected to provide a broad suite of characterization data for these nylon nanoplastics and TEM images are not included.   

-Figure 1D, the DLS data should be reported for Number% and Intensity.” 

  • Response from Authors: As requested, we have revised the size distribution graphs in Figure 1D to show Intensity vs. Hydrodynamic diameter.

-Gene expression should be investigated for proper conclusion.” 

  • Response from Authors: The authors appreciate this request to investigate gene expression. However, the collection of gene expression data cannot be completed given our resources. To ensure the awareness of the readership to this topic, we have included a statement that further studies are necessary to understand gene expression (please see lines 490-491).

-PS nanoparticles as control samples should have the same sizes of Nylon nanoparticles (based on TEM for both).” 

  • Response from Authors: The authors acknowledge that the smaller size ranges of PS nanoparticles used in this study (61 ± 14 nm; tested in deionized water) did not exactly match the nylon-11 NPs (127 ± 51 nm; tested in deionized water). The authors further revised the article to highlight the control sample was not exactly the same size to ensure the readership recognizes this concept (please see lines 409-411).

-A size-dependent study for at least 4 different sizes of Nylon is recommended.” 

  • Response from Authors: The authors appreciate this recommendation and enthusiastically agree that additional sizes of nylon would be extremely interesting to evaluate. Here, the formulations of nanoparticles that we developed in this manuscript showed two distinct sizes as presented in Table 1: Nylon-11 at 127 ± 51 and Nylon-6 at 465 ± 132. Further work is required to tune the processing parameters to achieve additional size distributions.

-Cytotoxicity should be compared with biocompatible nanocarriers that could be found in following references: International journal of pharmaceutics, 2019, 569, 118580; New Journal of Chemistry, 2018, 42 (12), 9690-9701; Journal of Drug Delivery Science and Technology, 2020, 57, 101715; Advanced Powder Technology, 2020, 31 (9), 4064-4071; Materials Science and Engineering: C, 2020, 109, 110597; Mol. Syst. Des. Eng., 2022, : DOI: 10.1039/d2me00024e

  • Response from Authors: The authors thank Reviewer #2 for this suggestion. The authors appreciate this suggestion and agree that cytotoxicity comparisons would be useful across materials belonging to the same class. Since this manuscript primarily addresses nanoplastics comprising commodity plastics and the references that the reviewer suggested are nanocarriers for drug delivery, drawing such parallels is outside the scope of this manuscript. The nylon NPs discussed in this manuscript have been compared to other nanoplastics published previously. We further highlighted the cytocompatible nanoplastics relevant to commodity plastics in the text (please see lines 482-487).
